# Impacts of Agriculture on the Environment and Soil Microbial Biodiversity

**DOI:** 10.3390/plants10112325

**Published:** 2021-10-28

**Authors:** Adoración Barros-Rodríguez, Pharada Rangseekaew, Krisana Lasudee, Wasu Pathom-aree, Maximino Manzanera

**Affiliations:** 1Department of Microbiology, Institute for Water Research, University of Granada, 18071 Granada, Spain; dorybarros@ugr.es; 2Doctor of Philosophy Program in Applied Microbiology (International Program), Faculty of Science, Chiang Mai University, Chiang Mai 50200, Thailand; bee.pharada@gmail.com; 3Graduate School, Chiang Mai University, Chiang Mai 50200, Thailand; 4Research Center of Excellence in Bioresources for Agriculture, Industry and Medicine, Department of Biology, Faculty of Science, Chiang Mai University, Chiang Mai 50200, Thailand; krisanaba@gmail.com (K.L.); wasu215793@gmail.com (W.P.-a.)

**Keywords:** biodiversity, agriculture, biostimulants, environmental threats

## Abstract

Agriculture represents an important mechanism in terms of reducing plant, animal, and microbial biodiversity and altering the environment. The pressure to cope with the increasing food demands of the human population has intensified the environmental impact, and alternative ways to produce food are required in order to minimize the decrease in biodiversity. Conventional agricultural practices, such as floods and irrigation systems; the removal of undesired vegetation by fires, tilling, and plowing; the use of herbicides, fertilizers, and pesticides; and the intensification of these practices over the last 50 years, have led to one of the most important environmental threats—a major loss of biodiversity. In this study, we review the impact that agriculture and its intensification have had on the environment and biodiversity since its invention. Moreover, we demonstrate how these impacts could be reduced through the use of microorganisms as biostimulants.

## 1. Introduction

Since the appearance of human beings 2.5 million years ago in East Africa, biodiversity has been declining. This decline was intensified with the invention of agriculture approximately 10,000 years ago, sharply reducing plants and animal biodiversity. In response to agricultural requirements, humans, through domestication, have caused animals and plants to evolve artificially [1]. Additionally, wild plants and animals in farming areas have been affected by the increase in food production; the diversity of these wild organisms has declined [2]. Growing domesticated species involves different cultivation conditions and techniques, such as the use of fertilizers, tillage, changes in land use, and the use of specific varieties of cultivated plants, all of which depend on the availability of natural resources and have resulted in a profound reduction in biodiversity [3]. The increase in food production was achieved as a result of these alterations. Owing to this increase, the human population also increased in number. Estimates suggest that since the appearance of humans, approximately 2.5 million years were needed to reach a population of between 1 and 10 million, which occurred before agriculture was invented. Thereafter, since the invention of agriculture, only 5000 years were needed to double that figure and reach a population of between 5 and 20 million [4,5]. This acceleration in population increase resulting from agriculture also implies that a continuous increase in agricultural production is required to meet the demands of the growing population. Therefore, a continuous increase in the area dedicated to agriculture is needed to cope with the growing food demands. Today, we estimate that the area dedicated to agriculture is in excess of 38% of the earth’s surface [6]. An alternative solution is agricultural intensification, i.e., the increase in agricultural production per unit area through different farming techniques (Figure 1). However, agricultural intensification results in an additional loss of biodiversity. Agricultural expansion and intensification are together recognized as the key drivers of biodiversity loss in the 21st century [7]. 

Previous studies analyzed the impact of different agricultural practices on soil microbial biodiversity. In this work, we explored the impact that traditional practices have and compared them with the practices associated with the Green Revolution. In addition, we proposed a series of alternative practices based on the use of safe microorganisms in order to promote more sustainable agricultural practices and greater conservation of soil microbial biodiversity.

## 2. The Impact of Traditional Agricultural Management and Techniques

### 2.1. Floods

The origin of agriculture initially occurred near large rivers, such as the Euphrates, Tigris, Indus, and Huang He rivers, and on their floodplains. These areas are now home to nearly 2.7 billion people [8,9]. These rivers were associated with arid or semi-arid areas, and they caused floods that enriched the surrounding areas with nutrients present in the sediments (mostly nitrogen and phosphorous but also potassium, magnesium, sulfur, and calcium), making them suitable for cultivation. The increase in soil nutrients from the sediments from river floods results in increased soil microbial respiration, biomass, and enzyme activity [10]. Changes in the amount and type of decomposable nutrients and altered nitrogen fluxes affect plant growth and soil microbial biomass and biodiversity [11,12]. Floods also affect oxygen availability and, thus, in theory, create the ideal anaerobic conditions for switching an aerobic microbial community into a dominant anaerobic one [13]. In general, the microbial biomass is reduced after a flooding event, as was determined by Wagner et al. (2015). On the basis of the study of fatty acids, they observed a higher proportion of Gram-positive bacteria than Gram-negative bacteria in response to the flooding effect [14]. However, in the short term, the increased concentration of nutrients depends on the plant diversity of the area and the increased availability of nitrogen as a result of the flood. In areas with low plant diversity, typical in agriculture environments, these effects are more pronounced and higher alterations in biomass and soil microbial activity are found, improving crop production [10]. 

### 2.2. Irrigation

After achieving cultivation on riverbanks, expanding agriculture required an irrigation system that supplied water to plants that were farther from the river. To this end, irrigation systems were used, involving channels, ditches, and various water transport systems connected to the nearest river. Irrigation systems also alter ecosystems in a similar manner to floods, although the effects of irrigation on plants were mitigated by the use of furrows or ridges that prevented water from covering the plants. Regions in which rainwater is insufficient for agriculture, such as arid and semi-arid regions, depend on irrigation systems, and these have a substantial impact on soil microbiota. According to various studies, the continued use of irrigation in the cold desert sagebrush steppe resulted in a loss of soil nutrients; for example, a loss of up to 16% of the stored carbon and approximately one-third of the NH_4_-N was reported, as compared with non-irrigated soils. These changes in the soil’s nutritional contents also affect the microbial (bacterial and fungal) community. Despite the nutrient reduction, microbial richness, evenness, and diversity generally increased with irrigation, most likely due to the increase in water availability [15]. 

Currently, various irrigation systems are associated with wastewater treatment plants, which result in an increase in the total nitrogen and organic matter in the soils. However, despite this increase in nutrients, the microbiota of these soils is reduced by the presence of certain pollutants, such as heavy metals, including mercury [16], and by the introduction of human-derived microorganisms.

### 2.3. Fires

After the success of agriculture on the margins of rivers, and after the invention of irrigation systems, agriculture was extended to zones previously occupied by forests. Large areas of forest were burned to cultivate domesticated plants and to create new grazing areas [17]. This technique is still in use today, with an estimated 7.2% of global forests having been lost since 2000. This is especially prevalent in tropical regions (mainly South America and Africa), where a 60% reduction in intact forest landscapes was observed between 2000 to 2013 [18]. This type of expansion also contributes to the loss of biodiversity. Specifically, fires cause an important loss of both plant and animal biodiversity, which profoundly affects the ecosystem. In addition, these fires generate changes in the chemical and physical properties of the soil, as a function of the fire severity and soil type [19]. These changes in the soil properties are caused by the heat generated, which induces chemical oxidation of soil organic matter, which, in turn, affects the microbial composition of the soil. This also occurs during wildfires [20]. Studies conducted in Mediterranean forests affected by fires showed that the fire had an impact on the bacterial communities involved in the nitrogen cycle, causing a loss of the diversity of the *nif*H gene, which codes for the enzyme nitrogenase reductase [21]. The effect on the nitrogen cycle depends on the type of plants present in the soil prior to the wildfire. Prendergast-Miller et al. demonstrated that ammonium is the dominant form of soluble nitrogen found in the soil when eucalyptus trees are burned. In contrast, nitrate becomes the dominant form of soluble nitrogen in the soil after the burning of pasture [22]. Alterations in the microorganisms involved in nitrogen metabolism were found in response to the altered nitrogen cycle. An increase in *Actinobacteria* and *Firmicutes* phyla in the soil was noted after a wildfire in a Mediterranean forest [23]. *Actinobacteria*, *Proteobacteria*, and *Firmicutes* phyla were detected after the wildfire in a eucalyptus forest [22]. *Proteobacteria*, *Acidobacteria*, and *Actinobacteria* phyla were identified after a wildfire in a northern boreal forest [24]. Numerous species of the *Actinobacteria* phylum have been described as having a high tolerance to abiotic stresses, such as heat and a lack of nutrients and drought; thus, it is normal to find this type of bacteria after a forest fire [25]. 

Fire is still used to remove slash, stubble, straw, and plant debris from previous harvests. This practice causes increased soil erosion, increases soil pH, causes loss of nutrients, such as carbon, nitrogen, and sulfur, and, in general, results in a decrease in soil quality [26]. The burning of straw residues also pollutes the atmosphere, as it produces greenhouse gas emissions (including CH_4_ and N_2_O) that exceed the Intergovernmental Panel on Climate Change (IPCC) default value [27]. Furthermore, this adversely affects the soil microbiota and, more specifically, plant growth-promoting rhizobacteria (PGPR)—a particular type of rhizobacteria from the area surrounding the root that improve plant growth [28]. 

### 2.4. Tilling and Plowing

Overturning the surface layers of the soil to eliminate weeds and other unwanted plants and seedbed preparation in the cropping area is an alternative to fire. This can be achieved by either tilling or plowing, both of which contribute to the aeration of the soil and mixing when fertilizers have been added. However, tillage and plowing also alter the biological and chemical characteristics of the soil and promote erosion by reducing soil moisture and organic matter contents [29,30]. Both techniques affect the population of earthworms (Lumbricidae) and soil microorganisms. Both earthworms and soil microorganisms have an important impact on the biological, chemical, and physical properties of the soil and affect its quality [31]. Earthworms represent the largest component of animal biomass in the soil, and they increase soil quality by improving its structure and increasing nutrient availability [32]. The control of plant pathogens, the production of PGPR, and the production of plant growth-regulating substances have been previously described [33]. According to a recent meta-analysis, earthworms contribute to nutrient availability by releasing their casts, which contain higher phosphorus (84%), nitrogen (24%) [34] and organic carbon (an average of over 40–48%) than bulk soil. However, conventional tilling and plowing techniques markedly reduce the earthworm population, with immature worms, which make up 76–90% of the population, being particularly affected [35,36]. The adverse impact that tilling and plowing have on earthworms depends on the soil texture and climate conditions.

Zuber and Villamil described the negative impact that tillage has on soil microbial biodiversity and enzyme activity using a meta-analysis based on 139 observations [37]. Loss of moisture, changes in temperature, alterations to soil microclimatic factors, and access to organic matter all influence microbial communities [38]. Soil organic matter dynamics profoundly depend on microbial abundance and diversity [39,40]. Navarro-Noya et al. described the alterations in the bacterial community caused by tillage, and more specifically, alteration in *Actinobacteria, Betaproteobacteria*, and *Gammaproteobacteria*. In this study, the proportion of *Betaproteobacteria* was correlated with electrolytic conductivity and clay content, while the proportion of *Gammaproteobacteria* was correlated with the total organic carbon [39].

### 2.5. Fertilizers

Another milestone in the history of agriculture is the use of fertilizers. These compounds, initially of organic nature, came from domesticated animal dung or bird guano. Soil concentrations of certain chemical elements increased with the addition of fertilizers, helping crop plant growth. The most common elements found in fertilizers are nitrogen, phosphorus, and potassium in different proportions. A lack of or scarcity of such elements in many soils limits the productivity of the plants. However, the increase in the production of certain plants through the addition of fertilizers results in a reduction in plant biodiversity in the treated area [41,42]. The reduction in plant biodiversity due to soil fertilization is normally associated with increases in aboveground production [43]. This phenomenon can be explained with three different theories: (i) *the light asymmetry theory*, which is related to the delayed growth of slower-growing plants [44].; (ii) *the total competition hypothesis*, which suggests a belowground competition in addition to the aboveground competition [45]; (iii) the *litter hypothesis*, which points to the fact that increased production of certain plants produces an increase in litter production of these species, inhibiting the germination of seeds from other species [46].

On the basis of a meta-analysis of 115 experiments, a reduction in biodiversity is observed with the addition of nitrogen. This reduction is more remarkable when NH_4_^+^ is used than when NO_3_ is used as a fertilizer. This loss of species richness due to nitrogen addition has been shown to be more significant in warmer environments, resulting in a greater loss of nitrogen-sensitive species, such as legumes and non-vascular plants [47]. In addition, several studies show a reduction in species richness after the addition of phosphorus and other nutrients as a fertilizer [47].

The reduction in plant biodiversity due to fertilizer addition also results in a reduction in the biodiversity of the soil microbiota (at both the bacterial and fungal levels) [48]. Moreover, the addition of these fertilizers affects carbon availability, pH, and soil osmolarity. Furthermore, it generates toxicity due to the presence of certain ions, resulting in a reduction in the abundance of *Acidobacteria* and *Nitrospirae* and a slight increase in *Actinobacteria* and *Firmicutes*. This reduction in microbial diversity in response to nitrogen addition is associated with a reduction in microbial biomass [49].

The traditional use of fertilizers of animal origin also alters the microbial composition. This is associated with both the way in which the elements are found and the incorporation of microorganisms from the intestine of the animals [50,51].

### 2.6. Herbicides and Other Pesticides

Herbicides are chemical molecules designed to reduce the growth of unwanted plants, and thus, they promote the productivity of the plants used in agriculture. Herbicides normally have different weed targets (see Beffa et al. (2019) for a recent review) [52] and have a high oxidative potential due to the high number of electronegative residues in their structure, including chlorine, phosphoric acid, hydroxide, oxygen, sulfonyl, amines, etc. This increased oxidative potential and other molecular interactions also affect non-target organisms, including other photosynthetic organisms, shredders, primary and secondary predators, and decomposers, including various soil microorganisms, resulting in a general increase in members of the *Actinobacteria* phylum and other herbicide-tolerant bacteria [53]. In addition, microbial communities are enriched with microorganisms with the metabolic machinery to degrade and consume such chemicals [54]. Agriculture also makes use of other chemical molecules to kill fungi, nematodes, insects, and rodents that affect food production. Again, such molecules alter the biodiversity of the area in which they are used and have a particularly profound effect on soil microorganisms [55,56].

### 2.7. Other Aspects of the Green Revolution

A huge increase in the use of chemical fertilizers, herbicides, and pesticides coincided with the use of non-renewable fuel-driven machinery, highly efficient controlled watering systems, and specific plant varieties with higher yields, all of which tripled global crop production from 1950 in a process termed the Green Revolution. However, the outstanding increase in crop production associated with the Green Revolution had serious environmental impacts. These impacts include a marked increase in greenhouse gas emissions, a threatening dependence on fossil fuels, conflicts associated with water use and sovereignty, soil salinization, a remarkable reduction in biodiversity, and substantial damage to human health and the environment [57]. These effects have intensified due to the increased demand for meat on a global scale and the production of crops for biofuel [58,59]. Many researchers point to global climate change, water eutrophication, and soil salinization as the main consequences of the Green Revolution, all of which cause tremendous biodiversity losses [60,61]. Therefore, alternatives to the current agriculture techniques associated with the Green Revolution are needed in order to prevent a massive reduction in biodiversity. An alternative method with which to tackle the increased food demand is internal ecosystem engineering, which can be seen as an alternative to the external manipulation of ecosystems associated with the Green Revolution. Brender et al. coined the term Underground Revolution to denote a method that provides the appropriate combination of organisms to the soil according to the plant requirements for growth [62]. 

## 3. Future Prospects and Concluding Remarks

Microorganisms can be used to promote plant growth by fixing atmospheric nitrogen and solubilizing inorganic phosphate; they can be used as alternatives to pesticides for the control of insects and other pathogens and can even promote plant growth in saline and arid soils [63,64,65]. The detrimental effects associated with various agricultural practices can be counterbalanced by the use of certain microorganisms. The use of biostimulants has been proposed as a method to reestablish the environment after floods and irrigation, fires, tilling, and plowing. Moreover, they are currently used as an alternative to chemical fertilizers, herbicides, and other pesticides. Biostimulants offer an environmentally friendly technique to reduce the damaging effects associated with these chemicals.

Several studies demonstrate that the addition of *Rhizophagus irregularis* or *Glomus mosseae*, both types of arbuscular mycorrhizal fungi (AMF), facilitates plant growth by enhancing the absorption of nutrient elements, such as phosphorous, and by promoting proline accumulation and improving root architecture under flooding conditions [66,67]. Furthermore, PGPB, such as *Pseudomonas fluorescens* REN_1_ and *Pseudomonas putida* UW4, have been used to protect plants such as *Rumex palustris* from floods owing to their ability to promote root elongation in plants through the activity of ACC deaminase and by reducing the production of indole-3-acetic acid (IAA) under constant flooded conditions [68,69]. In addition to microorganisms, other molecules, known as biostimulants, can promote plant growth by altering plant hormonal profiles [70]. Similar effects have been observed when applying various types of biostimulants based on fermented vegetable extracts and seaweed together with phytohormones. They provide protection for plants from floods but also improve soil microbial biodiversity by increasing the activity of soil enzymes involved in nitrogen fixation, phosphorus solubilization, the production of organic substances, and oxidation-reduction processes [71]. 

Soil toxicity resulting from fires used for agricultural practices can be reduced by AMF, PGPR, and other biostimulants. A recent study by Turjaman and Osaki (2021) described the use of AMF, ectomycorrhizal (ECM) fungi, and PGPR to restore soils affected by fires [72]. These strains reduce the toxicity of soils by transforming the pollutants into less toxic forms, regulating the bioavailability of certain molecules by producing certain chelators, and by releasing certain extracellular enzymes and hormones. Several PGPR belonging to the genera *Arthrobacter*, *Pseudomonas*, and *Bacillus* have been effectively used for the remediation of heavy metals, aromatic hydrocarbons, and to restore soil pH in soils affected by fires. Various bacteria also enhance the efficiency of bioremediation by promoting plant growth, alleviating pollutant phytotoxicity, improving ecosystem resilience, altering the bioavailability of pollutants in soil, and increasing translocation within plants [73,74]. Deforestation resulting from fires reduces evapotranspiration, which translates into increased aridity and desertification. The stress produced by water scarcity can be reduced by using certain microorganisms such as *Microbacterium* sp. 3J1 [75]. 

Other biostimulants have been used to reduce the detrimental effect associated with pollutants such as herbicides and pesticides. These dangerous chemicals can be replaced by biostimulants, which are widely used as biological control agents to antagonize and suppress destructive entomopathogens and bacterial and fungal pathogens in several ways [76]. The use of chemical fertilizers, herbicides, and pesticides also increases soil salinity, which can be alleviated by using certain microorganisms such as various species of the genus *Dermacoccus* [77]. 

It is important to guarantee that the introduction of biostimulants does not have a negative impact on the environment or human health. Vílchez et al. proposed a combination of bioassays to numerically determine the biosafety of bacterial strains that are intended to be released as biostimulants [78]. The European Union has recently regulated the use of bacteria as biostimulants in Regulation (EU) 2019/1009. However, the criteria established for shortlisting microbes that can be used are, in our opinion, inappropriate, as they are based solely on taxonomical criteria, as described by Barros-Rodríguez et al. [79]. In addition, appropriate assays should be incorporated to guarantee the efficiency of microbes used as biostimulants, including heat-inactivated microorganisms as controls, and these should be coupled with a proper analysis of the biostimulant expiration date [79]. 

In conclusion, the addition of certain biostimulants, in combination with appropriate management practices, would increase biodiversity in agricultural soils, providing specific functions and enhancing the overall ecosystem, which is of particular relevance in light of the increasing demand for food.

## Figures and Tables

**Figure 1 plants-10-02325-f001:**
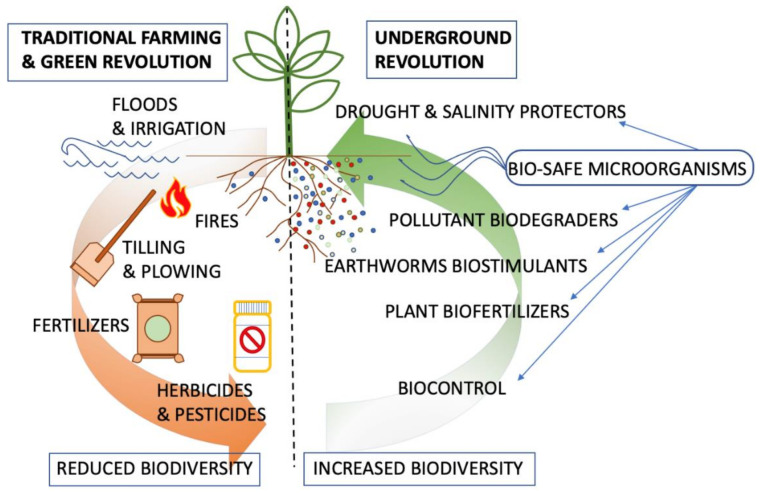
Conceptual diagram of traditional farming and Green Revolution, and Underground Revolution on soil microbial biodiversity. Colored dots represent microbial taxa.

## Data Availability

Data sharing does not apply to this article as no datasets were generated or analyzed during the current study.

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
