# Peer review of "Impacts of Agriculture on the Environment and Soil Microbial Biodiversity"

_plants, 2021, doi:10.3390/plants10112325_

Round 1

Reviewer 1 Report

Review Impacts of agriculture on environment and biodiversity

The review of the impacts of agriculture on the environment and biodiversity presents interesting aspects, especially in the effect of agriculture on the soil microbiota, the loss of soil quality, and the impact on the reduction of biodiversity.  Some aspects could be improved.

1.-The title of the review could be more specific because the review focuses more on biodiversity change in the soil microbiota.

2.- The work would be better structured. Could be:

1.Introduction.

2.Impact of management and techniques of traditional agriculture

3.Future prospects and Concluding remarks.

The sections number should be corrected

3.- Comments

Pg2 line 62 “Recent study has shown that river floods result in an increase in soil nutrients derived from the sediments based on an analysis of soil microbial respiration, biomass, nutrient limitation and enzyme activity (González Macé et al., 2016).”

Could you improve this sentence? The meaning is not clear.

Pg4 line 152 Meaning of term PGPR

Pg4 line 196 “However, studies regarding the reduction in species richness due to the addition of phosphorus and other nutrients as a fertilizer are inconsistent, some studies claim that there is such reduction (Fay et al., 2015), whereas other studies suggest an absence of such negative effect (Soons et al., 2017).

You might reconsider this statement. Throughout the paragraph "Fertilizers" arguments are given about the loss of biodiversity with fertilization. The bibliography that supports the lack of effects on biodiversity is insufficient, also, the work of Soon et al., (2017) does not support this affirmation.

Pg5 line 215 “Herbicides normally aim at different targets of weeds, such as acetyl-CoA carboxylase, acetolactate synthase, photosynthesis in Photosystem II, protoporphyrinogen, phytoene desaturase, 4-hydrophenyl-pyruvate-dioxigenase, carotenoid biosynthesis, 5-enolpy-ruvylshikimate-3-phosphate synthase, glutamine synthetase, dihydropteroate synthase, microtubule assembly, synthesis of very long chain fatty acid and synthesis of the cell wall”.

 I do not consider relevant this list of compounds that are affected by herbicides. It would be more interesting to indicate a review of the works describing these mechanisms.

Pg5 line 234 “Consequences of the Green Revolution”

This section could be redundant. The problems described in previous sections are a consequence of the green revolution. The name of the section could be changed: "other aspects of the green revolution"

Author Response

Dear Reviewer 1,

Thank you for your comments.

  1. With regards to your first concern, the title has been modified referring to the soil microbiota.
  2. The different sections have been renumbered, following your suggestions.
  3. With regards to your specific comments.
    1.  We have tried to improve the sentence in pg 2, line 62 for better understanding.
    2. We have tried to explain the term PGPR
    3. We have rephrased the information regarding the addition of phosphorous fertilizers to avoid misunderstanding  (the reference by Soons et al., 2017 has been removed).
    4. The list of plants compounds that are affected by herbicides has been removed and referred to a pertinent review.
    5. The section regarding the "Consequences of the Green Revolution" has been modified accordingly to your suggestions to "Other aspects of the Green Revolution"

We appreciate all your comments, since we believe they contribute to produce a much better article.

Sincerely,

Reviewer 2 Report

Research carried out by the author seems to be important  to the development and enhancement of existing information on this subject. The paper can be accepted for publication after the corrections have been made.

  1. Please be sure that a manuscript thoroughly establishes how this work is fundamentally novel. Specific comparisons should be made to previously published materials that have a similar purpose. Please present a strong case for how this work is a major advance. This needs to be done in the manuscript itself, not just in the response to review comments. This is a very important point in terms of which I will further consider the manuscript.
  2. Please be sure that the abstract and the conclusions section not only summarize the key findings of the work but also explain the specific ways in which this work fundamentally advances the field relative to prior literature.

       3. Words used in the title should not be repeated in the keywords.

  1. Please indicate the aim of the research.
  2. The chapter on the impact of fertilizers on soil biodiversity should be expanded.The authors should focus on the impact of both organic and mineral fertilization on soil biodiversity.

A novel information you can find in the following papers:

  • Ye, C., Huang, S., Sha, C., Wu, J., Cui, C., Su, J., ... & Xue, J. (2020). Changes of bacterial community in arable soil after short-term application of fresh manures and organic fertilizer. Environmental Technology, 1-11.
  • Grzyb, A., Wolna-Maruwka, A., & Niewiadomska, A. (2021). The Significance of Microbial Transformation of Nitrogen Compounds in the Light of Integrated Crop Management. Agronomy, 11(7), 1415.
  • Grzyb, A., Wolna-Maruwka, A., & Niewiadomska, A. (2020). Environmental factors affecting the mineralization of crop residues. Agronomy, 10(12), 1951.
  • Gautam, A., Sekaran, U., Guzman, J., Kovács, P., Hernandez, J. L. G., & Kumar, S. (2020). Responses of soil microbial community structure and enzymatic activities to long-term application of mineral fertilizer and beef manure. Environmental and Sustainability Indicators, 8, 100073.
  1. The authors analyzed mainly the issues of the impact of herbicides on soil microbial biodiversity. They should also focus on the effects of other pesticides on soil biodiversity.
  2. The length of the paper has to be increased significantly, increase also the detail of the conclusions.
  3. Make sure the references are added correctly according to the journal's instructions.
  4. The language correctness should be verified by a native speaker.

Author Response

Dear Reviewer 2,

We appreciate your comments. We have included a paragraph at the end of the Introduction, in order to make a strong case for how this work represents a major advance.

In addition, the English language has been checked by specialist.

We hope you consider the manuscript as appropriate for publication now.

With kind regards,

Reviewer 3 Report

Dear Authors,

I revised the manuscript "Impacts of agriculture on environment and biodiversity" submitted to Plants journal. The title of the manuscript and the abstract made me very interested, and I was expecting a very engaging, cross-cutting paper. Unfortunately, I was very disappointed. There is no clear structure to the manuscript and no connection between study sections. This manuscript lacks a clear vision and flow of ideas between sentences, paragraphs and sections. The review paper must describe the issues analyzed in broad terms. As a result of this analysis, the sections must be considerably longer. Also missing is an important chapter on livestock production, which is an essential element of agriculture and is fully dependent on it. In my opinion, it is good to include figures and tables in the manuscript to give a broader view of the issues being analyzed.

However, I have the following specific comments because the technical aspect of the manuscript needs substantial improvement:

  1. References should be numbered in order of appearance and indicated by a numeral or numerals in square brackets—e.g., [1] or [2,3], or [4–6].
  2. The same section number "4" is repeated five times.
  3. References should be numbered. In addition, the style of the references is not in accordance with the requirements of Plants. I suggest using a bibliography software package like Mendeley, Zotero, EndNote etc. 

Author Response

Dear Reviewer 3,

We regret your view on the manuscript. We understand that the title could mislead some readers, and therefore we have modified the title to focus on microbial biodiversity to avoid the same impression on other readers. 

We have used a proof reading service trying to facilitate the reading of the text.

Also, following your instructions, we have included a Figure, trying to summarize the main issues raised in the work.

1. With regards to the reference format, we will modify them if requested by you or the editor. However, we have followed the Instructions for Authors for Plants that now accepts free format submission, where "References may be in any style, provided that you use the consistent formatting throughout."

2. Sections numbers have been amended.

Thank you for your comments. 

We hope that you find this work appropriate for publication.

With kind regards,

Round 2

Reviewer 3 Report

Dear Authors,

I have reviewed the changes you have made to the manuscript. Unfortunately, the changes are minor. The title of the manuscript has been changed and reflects the nature of the paper a little better. However, I still believe that is no clear structure to the manuscript and no connection between study sections. This manuscript lacks a clear vision and flow of ideas between sentences, paragraphs and sections. The review paper must describe the issues analyzed in broad terms! As a result of this analysis, the sections must be considerably longer!

In my opinion, the manuscript in its current form is not suitable for publication in a highly impacted journal like Plants.

Author Response

Dear Editor,

We have followed the instructions from the three different Reviewers. We understand that Reviewer 3 does not like the style of the manuscript. However, to our opinion there is no clear and concrete instructions, given by this Reviewer, to improve this article other than making it much longer, improving the structure and connecting the different sections. To our opinion (and to the opinion of the other two Reviewers), the article has a clear connection between sentences, paragraphs and sections, and seems to be of enough quality for publication. Therefore, we understand that no additional issues must be described.

In consequence, we would appreciate if you could accept the manuscript as it is, given the work we have already done to improve the original manuscript originally sent.

With kind regards,

Maximino Manzanera.

Round 3

Reviewer 3 Report

Dear authors,
I accept your explanations, but I still have some doubts about some issues, in particular the length of the sections.
However, I have noted the improvements that have been made to the manuscript, and I believe that the Editor should take the final decision.